# Protocol for a theory-based, mixed methods evaluation of Cynnau|Ignite: an active learning programme to foster positive research culture through leadership development at Cardiff University

**Charlotte Hennah** [1]*, **Maleeha Rizwan**[2], **Nicola Edwards**[3], **Hayley Beckett**[4], **Michele Convery**[2], **Heidi Dawson**[2], **Jonathan Morris**[5], **Mair Rigby**[2], **James Vilares**[3], **Karin Wahl-Jorgensen**[6], **Rebecca Williams**[2], **Sofia Gameiro** [1]*

1 School of Psychology, Cardiff University, Wales, UK, 2 Organisational and Staff Development, Cardiff University, Wales, UK, 3 Strategy and Operations Team, The Research Service, Cardiff University, Wales, UK, 4 Human Resources, Cardiff University, Wales, UK, 5 School of Welsh, Cardiff University, Wales, UK, 6 School of Journalism, Media and Culture, Cardiff University, Wales, UK

* hennahc@cardiff.ac.uk (CH); GameiroS@cardiff.ac.uk (SG)

## Abstract

A recent UKRI call for training programmes to improve research culture at Higher Education Institutions led Cardiff University to design Cynnau|Ignite. This active learning programme intends to foster positive research culture practices through empowering staff towards leadership on this agenda, and by embedding learners in environments where these practices are highly valued. Cynnau|Ignite will be implemented within Cardiff University as three tailored programmes, dependant on job roles: Teaching and Research (64 learners), Research-Only (32 learners), and Professional Services, Technicians and Specialists (32 learners). This protocol details a theory-based, mixed methods evaluation of Cynnau|Ignite's effectiveness, acceptability, and feasibility as a tool to improve research culture practices and perceptions within Cardiff University. This knowledge will help inform future iterations of Cynnau|Ignite and similar initiatives, while advancing methods for the evaluation of positive research culture practices, and has potential to inform choices regarding funding and evaluation of research culture initiatives at other Higher Education Institutions. Here, we outline the Cynnau|Ignite Theory of Change used to inform our primary outcomes and describe our plan to evaluate the programme's effectiveness by measuring learners' self-reported positive research culture practices (intentions and behaviours) before and after taking part in Cynnau|Ignite, in comparison to a control group of staff who were not involved in the programme. Additionally, data monitoring, sessions observation, and semi-structured interviews will be conducted to establish programme acceptability and feasibility for stakeholders including learners, the delivery team, and university management. We hypothesise Cynnau|Ignite will be largely acceptable and effective in boosting learners' positive research culture intentions compared to a control group, but believe behavioural implementation may be adversely affected by common barriers

**Data availability statement:** No datasets were generated or analysed during the current study. All relevant data from this study will be made available upon study completion.

**Funding:** This study is funded by the Wellcome Trust [228090/Z/23/Z] https://wellcome.org. Funders were not involved in the study design, data collection, analysis, decision to publish or preparation of this protocol.

**Competing interests:** The authors have declared that no competing interests exist.

such as insufficient time and resources. Overall, we expect Cynnau|Ignite to improve short-term positive research culture practices and perceptions at Cardiff University, but its potential long-term impact is unclear.

## Study registration

Number: ISRCTN15575518

## Introduction

### Rationale

The Royal Society defines research culture as "the behaviours, values, expectations, attitudes and norms of our research communities. It influences researchers' career pathways, and determines the way that research is conducted and communicated" [1]. In recent years the positive research culture (PRC) agenda has gained traction worldwide, with Higher Education institutions (HEIs) actively pursuing equality, diversity and inclusion, while promoting positive behaviours such as open research, integrity and collegiality. Initial reviews discuss substantial problems for those working in research including a culture of overworking, low job security, and nonoptimal work conditions and practices (e.g., bullying and harassment) leading to poor mental-health and wellbeing among working staff [2]. Funders and other stakeholders are calling on the sector to address these issues and build a more positive research culture in which staff can thrive, and there is a growing realisation that failure to further this agenda will ultimately compromise both staff wellbeing and research output quality [3]. The rapidly accelerating drive to improve PRC is visible in its integration into the Research Excellence Framework's assessment of People, Culture and Environment in 2029 [3], which will pressure HEIs to make widespread changes in policy and practice if they want to be competitive for research funding.

One way the sector has tried to facilitate such changes is through educational programmes and initiatives. A recent UKRI review identified 292 initiatives, largely aimed at researchers and those supporting research activities, existed to promote positive research culture behaviours and values as they are noted in the UKRI PRC Framework [4]. However, this review also highlighted that the effectiveness of such initiatives is largely unknown due to a lack of rigorous evaluation [4], with the recommendation that future initiatives need to strengthen this component. In a period in which the Higher Education sector is struggling financially, and resources to foster PRC are likely to be diminish [5], it is crucial that the PRC agenda is furthered with an evidence-based approach.

In response to this shift, Cardiff University developed Cynnau|Ignite. This programme sits within the university's holistic plan to promote PRC, and is rooted in the belief that culture change cannot be forced through a top-down approach, but rather everyone can and should advocate for, and contribute to positive culture change. Cynnau|Ignite seeks to promote PRC by imparting the knowledge, skills, and leadership development necessary for learners to feel empowered to advocate for a more positive research culture. The programme will be piloted at Cardiff University during AY 2024/2025, and will incorporate theory-based evaluation into every area of the programme's design and delivery to ensure rigour. Its design was informed by evidence showing that similar programmes, often using face-to-face training and team development, have been an effective method of achieving culture change within higher education [6] especially when supported by senior or management staff and appropriate resources and skills were imparted [7]. With the backing of university management,

Cynnau|Ignite aims to test the strength of advocating for PRC as route for change, via developing positive attitudes, cultural norms, skills, and leadership potential in Teaching and Research (T&R), Research-only (R), and Professional Services staff, Technicians, and Specialists (PST/S) from all disciplines across the university, enabling them to enact and advocate for PRC.

Cynnau|Ignite encompasses a holistic multidimensional approach to PRC and consists of three streams including Research Culture, Research Skills, and Action Learning. The delivery of these streams will be tailored to T&R, R, and PST/S learners based on their role requirements. By fostering community engagement around PRC, the programme is expected to maximize holistic benefits for all learners and manifest in intention formation and behavioural enaction of PRC, as defined by the UKRI PRC Framework [4], leading to positive culture change across the institution [8]. Alongside this programme, a dedicated evaluation will investigate the acceptability, implementation, and effectiveness of Cynnau|Ignite and its ability to help foster PRC practices, empowerment and leadership development at Cardiff University. This evaluation will not only inform future iterations of Cynnau|Ignite at Cardiff University, but may also lead to a greater understanding of the effectiveness of PRC initiatives in the UK, and how HEIs could most appropriately invest resources to promote these changes.

Cynnau|Ignite's evaluation, and elements of its design, are partially informed by Ajzen's Theory of Planned Behaviour (TPB) [9]. While there is not yet a theory of organisational behaviour and culture change that fully underpins all the programme aims to achieve, literature review shows potential for TPB to inform and evaluate organizational change [10]. A meta-analysis by Sheeran et al. [11] demonstrated considerable experimental support for TPB constructs and their effectiveness in predicting health behaviour change, as well as organisational change, where employee attitudes have long been proven critical for the success of interventions [12,13]. According to the TPB, colleagues' intentions to engage and champion positive RC practices and behaviours will be stronger if they have favourable attitudes towards PRC, perceive that significant others (e.g., colleagues, line-managers, senior management) expect them to engage in PRC (norms), and think they have the skills and resources to enact PRC (i.e., perceived behaviour control, PBC). Meta-analytic synthesis shows that such intentions are moderately predictive of actual behaviours [14]. Given its robust framework, comprehensive measurable constructs (attitudes, norms, control and intentions), and moderate predictive power, with intentions explaining approximately 22% of variance in behaviour [14] the TPB will be used in conjunction with the UKRI PRC Framework (which also encompasses an holistic and comprehensive conceptualisation of PRC) [4], to map elements of the Cynnau|Ignite programme design and evaluation.

## Objectives

The primary aim is to reach a conclusion about the impact of Cynnau|Ignite on PRC at Cardiff University. Underpinned by theory-based approaches and a mixed-methods multi-informants design, evaluation will test whether Cynnau|Ignite can be implemented as planned, investigate how learners and other stakeholders (Heads of School/line managers, senior management and delivery team) will engage with it, and measure change in learners' PRC intentions and behaviours. Evaluation will address the following research questions:

1. How do relevant stakeholders (e.g., design and delivery team, learners, heads of school/line managers, senior management, advisory board) understand Cynnau|Ignite, the outcomes it creates, how, and in what contexts?

2. How do relevant stakeholders evaluate Cynnau|Ignite and the outcomes it creates, and what factors (learners, organizational, programme) shape its perceived value?

3. To what extent is it feasible to implement Cynnau|Ignite at Cardiff University and what are barriers and facilitators (learners, organizational, programme) to its implementation and adoption?

4. To what extent is Cynnau|Ignite effective in promoting learners' intentions and behavioural enaction of PRC practices at Cardiff University; and improving stakeholders' perceptions of and organisational indicators of PRC at Cardiff University.

Based on these questions, we hypothesise:

1. Stakeholders will be able to describe, and in many cases may have helped define, the programme's purpose, and how it seeks to create change. Expectations of the overall cost/benefit will differ based on perspective (e.g., Heads of School and line managers may perceive greater disruption resulting from Cynnau|Ignite, compared to the delivery team).

2. All stakeholders will value the programme, while understanding that the learners' ability to translate intentions to engage with PRC into actual behaviours will depend on barriers and facilitators they experienced throughout.

3. Implementing Cynnau|Ignite at Cardiff University will be feasible and practical due to its bespoke nature and close alignment with university values. We expect that immersing learners in an environment which directly fosters Cardiff University's PRC aims, with support from senior management, will facilitate adoption of the programme. Additionally, the iterative, adaptable and bespoke nature of this programme may increase the feasibility of its wider adoption, and potential implementation in other institutions. However, we predict it's high-resource design will make implementation and adoption vulnerable to common barriers such as lack of time, financial and human resources.

4. Cynnau|Ignite will be acceptable and effective in increasing learners' intentions of PRC practices at Cardiff University, compared to a control group of staff who did not participate in Cynnau|Ignite. We expect this difference to be evident across multiple dimensions of the UKRI PRC framework targeted (specifically research integrity, open approach to research, communicating research, ensuring an inclusive work environment, building collegiality, and realising impact [4]). Finally, we expect learners' enaction of PRC practices will have a ripple effect at organisational level, improving perceptions of and indicators of PRC at Cardiff University.

## Cynnau|ignite programme

Cynnau|Ignite consists of three streams, as depicted in Fig 1: Research Culture—aiming to foster positive attitudes and commitment towards PRC, Research Skills—helping to impart the knowledge and skills necessary to action and advocate for PRC, and Action Learning - supporting learners in developing bespoke practice projects or small award grant applications, with potential funding of up to £2000 to facilitate individual projects around the theme of positive research culture.

Three versions of the programme will be delivered, tailored to the career needs of Early Career Researchers (ECRs) on Teaching and Research (**T&R**: 2 cohorts of 32 learners), or Research-only (**R**: 1 cohort of 32 learners) career pathways, as well as Professional Services colleagues, including Technicians and Specialists (**PST/S**: 1 cohort of 32 learners) across all disciplines in the university. Details of each tailored programme can be seen in Fig 1.

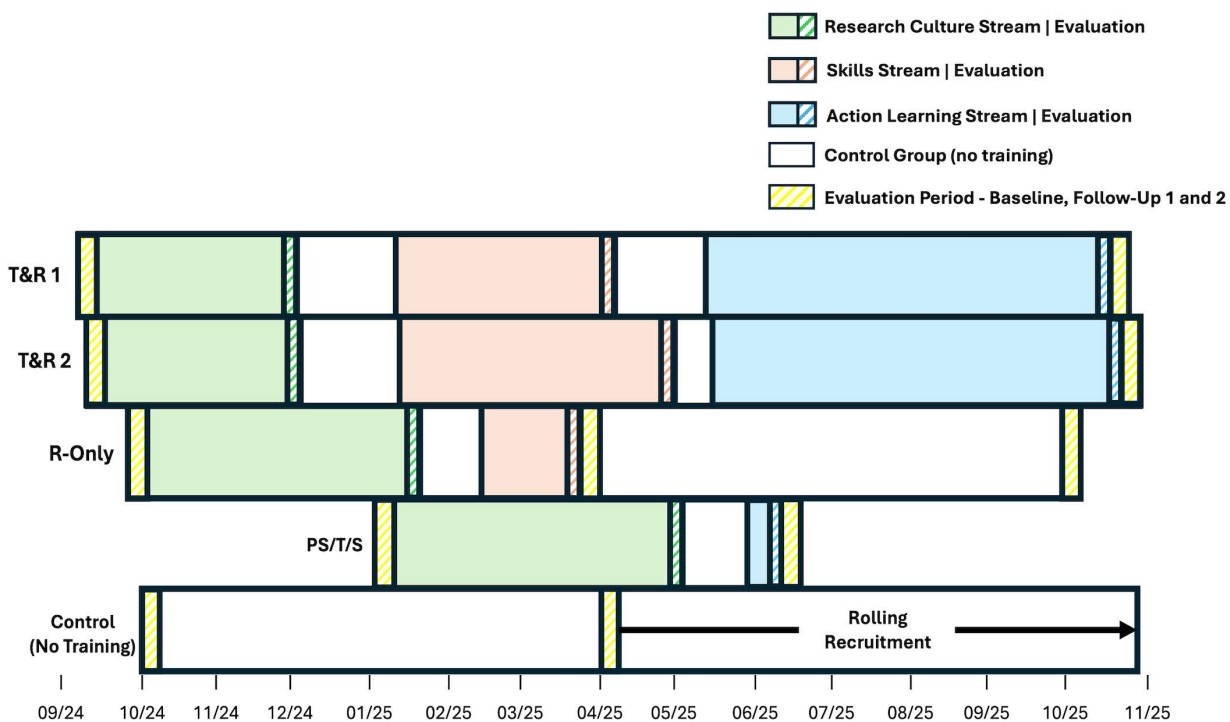

**Fig 1. Schematic chronologic representation of Cynnau |**Ignite's implementation and evaluation.

Fig 2 presents Cynnau|Ignite's streams, modules, and sessions as mapped against the dimensions of the UKRI PRC framework that are targeted by the programme. Module Learning Outcomes can be found in supplementary material.

The programme's tailored training sessions are hypothesized to help learners promote positive values and attitudes towards PRC, develop a relevant skill set (research career development, leadership) leading to higher perceived efficacy in enacting PRC, and aid in the creation of a community of practice that normalises PRC. In turn, according to the Theory of Planned Behaviour these elements should lead to the creation of strong intentions to practice, and advocate for, PRC and its subsequent behavioural enaction, which is supported by the Action Learning stream of the programme. Learners' enaction of PRC practices is also expected to lead to improvements in stakeholders' perceptions of, and in organisational indicators of, PRC at Cardiff University.

Cynnau|Ignite will be run as an interactive programme and all members of the delivery team have experience with participant engagement and a background in designing and implementing training programs for leadership, research skills and/or culture. Except for the Influencing Change module, all content was designed by a network of internal Cardiff University specialists aided by contributors such as specialists or Heads of School, who have specific knowledge and skills pertaining to the session they have been asked to lead. However, Influencing Change will be delivered by an external provider, 64 Million Artists, due to their expertise in creative leadership and helping academics to raise their research ambitions and shape thriving university environments. This provider was able to take a part of their existing "Leading Researchers" programme and create bespoke content to meet the requirements of Cynnau|Ignite's Influencing Change learning outcomes. Discussions between both parties demonstrated similarities in leadership ethos and partnering expectations, for example,

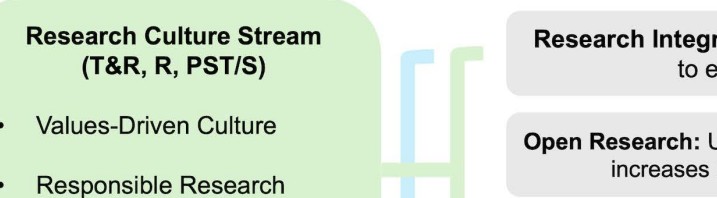

**Fig 2. Cynnau|Ignite streams and modules, mapped against relevant dimensions of the UKRI Research Culture Framework.**

enthusiasm about creativity, belonging, and meaningful connections. Compared to other external providers, 64 Million Artists were able to provide additional strengths mapping opportunities, as well as make all session dates within the budget expected.

The programme will primarily be delivered face-to-face, however online access will be provided if it is requested as a reasonable adjustment due to personal circumstances (considered on a case-by-case basis). All training sessions will be held in conference rooms, either within Cardiff University or at an external venue, and rooms will be large enough to accommodate 32 participants and several members of the delivery team. Each session will run from 9 am until 4.30 pm, inclusive of lunch and rest breaks, and all learners must commit to attend all modules during the application process, with some flexibility given for those who cannot attend a single date. Attendance will be monitored and recorded.

Throughout, learners will have access to a physical Cynnau|Ignite workbook, featuring materials which guide them through all individual and group activities for each module (e.g., values and self-leadership assessments), and an online Microsoft Teams channel, where learners will have on-demand access to PowerPoint slides summarizing each module, and related reading materials. Using this channel, learners will have the ability to interact with

programme facilitators and others in their cohort and share PRC resources. Learners are not restricted from engaging in other PRC initiatives or training programmes on their own volition, both within and outside Cardiff University.

## Study design

This is a pre-registered (ISRCTN15575518), theory based, mixed-methods and multi-informants' prospective pragmatic evaluation of Cynnau|Ignite. A pragmatic approach fits this evaluation because we aim to provide definite answers to high-priority (but not exhaustive) questions pertaining the Cynnau|Ignite programme, when it is being flexibly implemented at Cardiff University with little selection of learners and low bias control, and with limited resources for evaluation [15]. The evaluation is underpinned by the UKRI PRC Framework [4] and a Cynnau|Ignite Theory of Change, presented in Fig 3, which demonstrates how the programme is intended to work and the assumptions behind this theory. This study will consist a process evaluation to investigate Research Questions 1–3 (as described above), and a quantitative evaluation of effectiveness to investigate Research Question 4.

The effectiveness quantitative evaluation (displayed alongside Cynnau|Ignite's implementation plan in Fig 1) is a prospective evaluation with three assessment moments: within two weeks of the start of the programme (baseline – T1), and two weeks following the final session

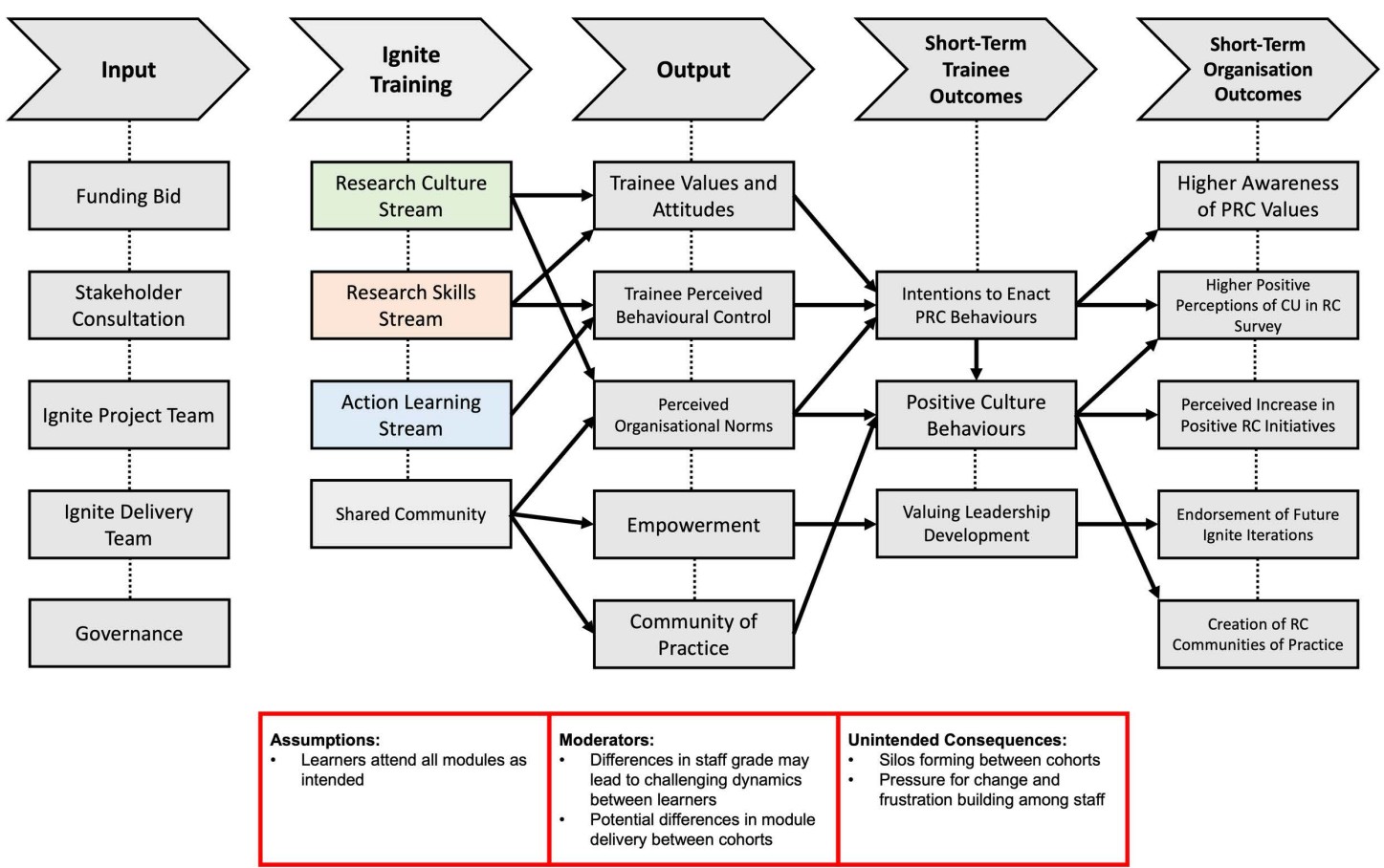

**Fig 3. Cynnau |Ignite Theory of Change describing inputs, activities and anticipated short-term outcomes.** Potential Moderators and Unintended Consequences are outlined in red.

(short-term follow up – T2), then six months after the end of the programme (mid-term follow up – T3). Fig 4 presents the Participant Flowchart. All cohorts will complete the baseline and short-term follow up, with an additional mid-term follow up for the R-only cohort (as course timetabling allows). A control group of university staff who did not take part in Cynnau|Ignite will fill in the baseline and short-term follow up with a 6-month gap in between, which equals the shorter period of Cynnau|Ignite implementation across the three groups (i.e., for the PST/S group) and is therefore comparable with the baseline and short-term follow up assessments for the cohorts.

## Outcomes

The study outcomes are presented in Table 1, organized per the study's research questions and Bowen's framework [16] which evaluates different areas of intervention implementation that are directly relevant to our research aims.

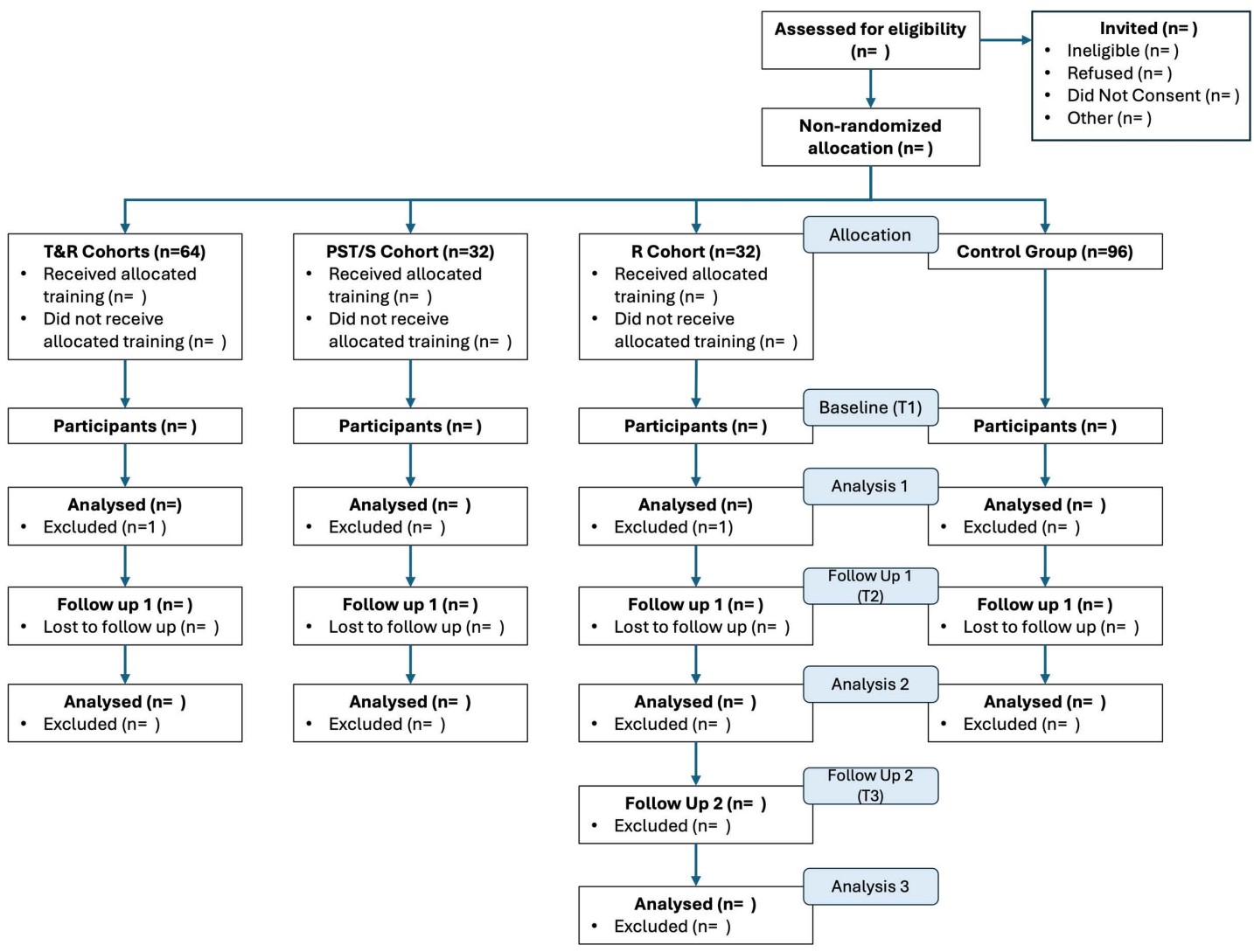

**Fig 4. Participant Flow Chart according to CONSORT guidelines (2010).**

**Table 1. Cynnau|Ignite's evaluation outcomes (organised by research question), methods and informants for evidence gathering.**

| | Dimension evaluated* | Outcomes | Informants | | |
|---|---|---|---|---|---|
| | | | Learners | Stakeholders | Delivery Team |
| Research Q1 | **Acceptability** How well is Cynnau|Ignite received? | Expected (positive and negative) outcomes of attending themes and overall participation in Cynnau|Ignite, for learners and their school/unit | PE | PE | PE |
| | | Perceived (positive and negative) outcomes (and cost-benefit) of attending themes and overall participation in Ignite, for learners and their school/unit | PE | PE | PE |
| | **Integration** How well can Cynnau|Ignite be integrated into Cardiff University? | Perceived disruption(s) caused by Cynnau|Ignite | PE | PE | PE |
| | **Effectiveness** To what extent is Cynnau|Ignite successful in achieving its goals? | Perceptions about what in Ignite contributed to outcomes perceived and how | PE | PE | PE |
| | | Perception about for who and under what circumstances does Ignite contribute (or not) to outcomes | PE | PE | PE |
| | **Impact** The short- and longer-term effects or influence of Cynnau|Ignite | Perceptions about long-term impact(s) of staff attending Ignite (e.g., career development) at personal, school, and Institutional level | PE | PE | PE |
| | | Perceived impact from and plans to continue/extend Action Learning projects (e.g., other contexts, further developments, networks, etc.) | PE | PE | PE |
| Research Q2 | **Demand** To what extent is Cynnau|Ignite likely to be used? | Number of brochures requested | DM | | |
| | | Number of EoI submitted | DM | | |
| | | Final number of participants & per/school | DM | | |
| | | Background and professional profile of applicants | PE | | |
| | | Perceptions of demand | PE | PE | PE |
| | | Attendance rate | DM | | |
| | | Attendance rate for optional Action Learning module (R) | DM | | |
| | | Contributors and learners' engagement during sessions | OBS | | |
| | **Acceptability** How well Cynnau|Ignite is received? Is it suitable and appropriate to meet the needs of participants, stakeholders and the delivery team? | General views of and feedback about Cynnau|Ignite | PE | PE | |
| | | Perceptions of most and least useful elements or sessions of Ignite | PE | | PE |
| | | For each Stream: multiple Likert-type questions assessing acceptability of content and delivery, plus one open-ended question for additional feedback | SE | | |
| | | Observation of and feedback/comments from participants re acceptability during sessions | OBS | | |
| Research Q3 | **Implementation** To what extent can Cynnau|Ignite be delivered as planned? | Perceptions of barriers and facilitators of programme implementation | | | PE |
| | | Description of changes to planned implementation and rational | | | PE |
| | | Description of changes to planned implementation and rational | | | DM |
| | | Description of changes happening during sessions and rational | OBS | | |
| | **Practicality** Can Cynnau|Ignite be carried out as planned with the resources available? | Perceptions of barriers and facilitators of attendance | PE | PE | PE |
| | | Perceptions of barriers and facilitators of implementing Action Learning projects | PE | PE | PE |
| | | Perceptions of barriers and facilitators of general enactment of Cynnau|Ignite learning, skills, and values | PE | PE | PE |
| | | Observation of and feedback/comments from participants re practicality during sessions | OBS | | |
| | **Adaptation** How well can we expect Cynnau|Ignite to perform at other HEIs or staff career development levels? | Perceived challenges of implementing Ignite at other HEIs or staff career development levels | PE | | PE |
| | | Perceived changes needed to implement Ignite at other HEIs or staff career development levels | PE | PE | PE |
| | **Integration** How well can Cynnau|Ignite be integrated into Cardiff University? | Perceived organizational fit of Cynnau|Ignite (resources, culture, practicalities…) | PE | PE | PE |
| | | Perceptions of sustainability of implementing Cynnau|Ignite | PE | PE | PE |
| | | Observation of and feedback/comments from participants re fit of Ignite with organizational environment/procedures during sessions | OBS | | |

*(Continued)*

**Table 1.** (Continued)

| | Dimension evaluated* | Outcomes | Informants | | |
|---|---|---|---|---|---|
| | | | Learners | Stakeholders | Delivery Team |
| Research Q4 | **Effectiveness**<br>To what extent is Cynnau\|Ignite successful in achieving its goals? | Analyses on TPB outcomes (intentions, attitudes, norms, perceived behaviour control) measured at T1 and T2. | QE | | |
| | | Differences in TPB outcomes (intentions, attitudes, norms, perceived behaviour control) for R-only learners, measured at T2 and T3 | QE | | |
| | **Impact**<br>The short- and longer-term effects or influence of Cynnau\|Ignite | Differences in perceptions of PRC at Cardiff University, as measured in the Research Culture Survey, between all Ignite learners and a matched group of CU staff who did not participate in Ignite, and between learners and control group. | QE | | |
| | | Nr of Action Learning projects implemented and impact narratives from projects | DM | | |
| | | Differences in uptake of mandatory training between learners and control group | DM | | |
| | | Institutional decision about funding Ignite longer-term | | DM | |

Legend. DM = data monitoring via University/Ignite procedures, OBS = semi-structured behavioural observations during Cynnau|Ignite sessions, QE = Quantitative evaluation conducted via online survey, PE = Process evaluation, based on individual semi-structured interviews or focus groups.

## Materials and methods

### Study setting

Cynnau|Ignite will be fully implemented at Cardiff University. The university is part of the Russell Group, comprised of 24 research-intensive institutions in the UK, and 90% of its research was classified as internationally excellent.

### Participants

The intervention group will consist of all learners, comprised of four cohorts (two T&R cohorts, one R and one PST/S cohort) with 32 learners in each. All those accepted into Cynnau|Ignite will be eligible to take part in the study and will integrate the intervention trainee groups. Selection criteria for Cynnau|Ignite will be those employed at Cardiff University at grade six, seven or eight as Teaching and Research, and Research-only staff, along with Professional Service staff, or Technicians at grades 5 to 8, in any discipline. Applicants will be expected to demonstrate that they have an interest in PRC and are available to attend all sessions, with allowances made for those who are unable to attend a single session. These criteria mirror common practice for similar training programmes at Cardiff University and will be likely to remain unchanged, unless if demand will need to be addressed in future Cynnau|Ignite iterations. We expect high participation and low attrition in the intervention group. No concealment or blinding will be used in this research. All learners will be allocated to either the Teaching and Research, Research-Only, or Professional Services, Technicians and Specialists version depending on their career path.

Inclusion criteria for the control group will be being a Cardiff University employee at grade six, seven, or eight who is not enrolled in the programme. Based on moderate attrition rates between 20–50% [17,18] observed in repeated measures surveys and organizational research, we intend to recruit a total of 96 staff members for this group, with a maximum of 32 participants each from T&R, R and PST/S roles. A power analysis conducted using GPower [19] indicates that this will provide enough power to detect small to moderate effect sizes (repeated measures ANOVA, p = .90, f = 0.15, $\alpha$ = .05) in differences between the learner and control groups in changes in intentions from baseline (T1) to follow up (T2) and in differences between the R and control groups in changes in intentions from baseline (T1) to follow ups (T2 & T3).

To assess the acceptability of Cynnau|Ignite, and its feasibility for use at Cardiff University as a tool to promote positive research culture and leadership development, we will invite five members from each of the intervention groups, as well as Heads of School and Directors of Research, and members of the delivery team to participate in the process evaluation via semi-structured interviews or focus groups. This sample size was deemed appropriate to support the in-depth exploratory qualitative analysis required while capturing data from all necessary groups involved in this research through purposeful sampling.

## Recruitment

The Cynnau|Ignite programme brochure will inform staff they will be invited to participate in this evaluation study. Staff who apply and are selected for the programme will receive an invitation to participate in the research in their Welcome Pack. In the event Cynnau|Ignite exceeds recruitment targets eligible participants will be chosen to maximise representation across roles, schools and units, however if this is not possible, participants will be randomly selected. As such, recruitment for the Cynnau|Ignite participant group will begin on 1st September 2024, and end 8th January 2025 (dependant on cohort start date). Recruitment for the control group will begin 1st September 2024 and run until 1st July 2025 or until target numbers have been reached and will first target staff who indicates an interest in attending Cynnau|Ignite (e.g., request brochure) but do not apply or are not selected, via email invitation. If recruitment targets are not reached with this strategy, participants will be recruited more widely within the university using CU communication channels. All non-trainee participants (e.g., heads of schools) will be invited to participate via email.

All prospective participants will be signposted to the Participant Information Sheet and Consent Form, where they will be required to provide full written informed consent before taking part in this research. After providing consent, participants will be directed to the baseline survey.

## Materials

Data collection materials are outlined below. For a detailed description, please see the 'Cynnau|Ignite Evaluation Materials Table' (S2 in supplementary materials), and the materials will be deposited in the UK Data Service repository at the end of the study (https://ukdataservice.ac.uk). All surveys in this study will be presented on Qualtrics (https://www.qualtrics.com). The quantitative evaluation, including the baseline (T1), and follow up (T2, T3) surveys will collect the following data:

- Participants' background (e.g., gender, ethnicity) and professional status (e.g., career path, type of contract).

- Participants' attitudes, subjective norms, PBC, intentions, and actual behaviour relative to PRC practices, organised according to the 6 dimensions of the UKRI PRC framework [4] that are targeted by Cynnau|Ignite (research integrity, open approach to research, communicating research, ensuring inclusive work environments, building collegiality, and realising impact) and evaluated with questions developed specifically for the study following the guidelines provided in the TPB manual [20].

- Research skills targeted by Cynnau|Ignite, with bespoke questions around creativity, leadership, and career development mapped to domains of the UKRI PRC Framework [4].

- Experiences at work, including participants' perceptions of a workplace community of practice (as defined by Harvard Business Review) [21], workload, and job satisfaction [22].

- For the follow up surveys only: One closed and one open question about uptake of other PRC initiatives or training during the Ignite|Cynnau training period, created for this study to mitigate the risk of other PRC initiatives impacting these results.

In addition to the primary quantitative evaluation surveys, short stream evaluations will take place during the last module of each stream. Learners will fill in short evaluation forms, approximately five minutes in length, presented on Qualtrics. Adapted from Rowbottom et al. [23] these consist of five-point Likert scales asking learners to reflect on their experiences throughout the stream, for example whether they found the content appropriate, useful, or boring.

Structured observations will also be used to investigate learners' engagement with, and reactions to, the programme content. A total of 12 half-day Cynnau|Ignite sessions, split between am and pm sessions, were randomly selected for observation (approximately three per cohort). Using a modified version of Lane and Harris' Tool for Measuring Student Behavioural Engagement [24] which we tailored to ensure the tool sufficiently captured the behaviour of university staff in an active-learning programme, we will note the number of learners exhibiting specific behaviours such as reading, writing, and making eye contact with the facilitator, at ten-minute intervals throughout observation sessions. This data will be noted anonymously, and at group level, along with narrative annotations where necessary for context.

Process evaluation data will be collected via semi-structured interviews and based on realist principles [25,26]. We aim to collect participants' views about the outputs (positive, negative) triggered by Cynnau|Ignite and how (theory of change), for whom, and under which circumstances these are triggered [27,28]. Finally, impact evaluation data will be collected via desk review and the above-mentioned interviews, and overall perceptions of PRC at Cardiff University will be measured in the biennial Research Culture Survey. (https://www.cardiff.ac.uk/documents/2741029-research-culture-survey-report-2023-executive-summary), and compared between the intervention and control groups.

## Analytical methods

Mixed methods analysis will be carried out, using SPSS and NVivo for quantitative and qualitative analysis respectively. Descriptive (n, proportions) and inferential (ANOVA, Chi-Square) statistics will be used to describe participants' background and professional characteristics. Research Questions 1–3 will be investigated using semi-structured interviews, as part of a process evaluation based on realist principles [24,25], which will cover the topics described in Table 1 using an explanatory sequential mixed methods design. Here, data monitoring, session observations and short stream evaluations (completed by learners at the end of each Cynnau|Ignite stream) will be quantitatively analysed using descriptive (n, proportions) and inferential (ANOVA, Chi-Square) statistics; these results will help to inform part of the process evaluation and subsequent qualitative analysis of semi-structured interviews. Content analysis with hybrid deductive and inductive coding [29] will be used to create an iterative framework based on existing literature, and any relevant data from our acceptability or effectiveness analysis. Due to the subjective nature of qualitative data analysis, we will take additional steps to ensure rigour and authenticity in these elements of our research including use of multiple data coders. At regular intervals, coders will be engaged, debriefed and encouraged to reflect on how their own biases may impact the analysis.

Effectiveness and impact (Research Question 4) will be reported for all learners, but the analytical strategy will depend on the final sample achieved. Cardiff University has a total of 26 academic schools, all of which are eligible for Cynnau|Ignite. To account for

non-independence of data we will use multi-level modelling [30] with learners (level 1) nested within schools (level 2), but only if the recommended 20 schools/units are achieved [31]. If this is not possible, we will use two-way Mixed ANOVA. Analysis will investigate how Cynnau|Ignite affects self-reported intentions and actual PRC behaviours (primary outcomes), attitudes, norms and PBC (secondary outcomes), with Group (Cynnau|Ignite, Control) as the between-subject variable and Time (T1, T2) as the within-subject factor. Similar analysis will be used to compare each Cynnau|Ignite group (T&R, R, PST/S) with the control group and to investigate changes in the study outcomes across a 6-month period following Cynnau|Ignite, the latter with Time (T2, T3) as the within subject factor for the R-only cohort. Statistical significance will be indicated if $p < 0.05$ and effect sizes will be reported for all analysis. Finally, we will quantitatively analyse the organisational impact of Cynnau|Ignite by investigating learner and control group differences with multilevel modelling, or Mixed ANOVA if the necessary units are not reached, in perceptions of PRC at Cardiff University. Descriptive and inferential statistics will be used to map the impact of Cynnau|Ignite on impact indicators, such as the implementation of Action Learning projects related to research culture, and learners' uptake of mandatory staff training compared to those who did not participate in Cynnau|Ignite.

### Research ethics approval

This protocol has been approved by Cardiff University's School of Psychology Research Ethics Committee (REF EC.24.06.11.7023G). Any amendments will be submitted to the ethics committee for review as necessary and highlighted in publications and/or dissemination of results.

## Discussion

This protocol describes a theory-based, mixed methods evaluation of Cynnau|Ignite's effectiveness, acceptability, and feasibility as a programme to promote PRC practices and leadership development. We hope this evaluation will provide insight into the usefulness of Cynnau|Ignite in its current form, the UKRI framework in describing research culture behaviours, and TPB's feasibility to evaluate these behaviours [4,9]. Knowledge gained from this study will be critical to pave the way forward for research culture initiatives and funding at Cardiff University and other HEIs, which is expected to be of growing significance throughout the sector.

First, results will inform about how acceptable this holistic leadership programme is to higher education staff in different career paths and the feasibility of implementing it within the sector. Knowledge will inform if bottom-up approaches to foster PRC via leadership development are an avenue for change worth pursuing, will shape future leadership and PRC programmes in an evidence-based way, and identify areas and didactics of PRC training that are particularly valued.

Second, results will clarify Cynnau|Ignite's effectiveness in promoting PRC values and practices. A major benefit of Cynnau|Ignite is its tailored approach to staff on different career paths, and our process evaluation should help to clarify the causal mechanisms through which Cynnau|Ignite operates, as well as the moderating role of context, and this could be instrumental to inform HEIs strategic decisions and policy intended to foster PRC.

Third, this study will provide comprehensive data about the impact of Cynnau|Ignite at organizational level, therefore directly reliably addressing the fundamental question of if, which, and how systemic change in RC can be achieved. Unfortunately, the time-limited nature of this funded project does not allow us to examine the sustainability of this change beyond 6-months (if any), nor does it allow us to capture the full extent of changes which may only manifest in the long-term (ripple effects), as is often the case in complex organisations

like Cardiff University. However, evaluation findings will increase HEIs and funders' understanding of how best to invest in PRC (especially in a context of reduced funding), by contributing insights into what are the most effective contributions of the programme, what are common barriers to and how do these obstruct change in PRC, how to scale up Cynnau|Ignite or similar initiatives to maximize organizational impact, and how to measure this in the longer term.

A fourth contribution of this project is advancing meta-research in higher education, by testing the application of the TPB [9] in this context and its integration with the UKRI research culture framework that forms the basis of our evaluation. While the UKRI framework constitutes significant conceptual advance in the definition of PRC [4], our TPB-based assessment constitutes a proposal for operationalising this holistic concept in a way that can be used to measure change.

As this is a pragmatic trial that has been retrospectively fitted around Cynnau|Ignite's design, there are some limitations. While the programme is tailored to specific groups of staff at Cardiff University, this tailoring is largely achieved on the basis of omission. T&R learners will complete the full training program, but those in R or PST/S groups will receive a truncated version. This complicates the study as groups will not receive equal amounts of training (i.e., active components will vary), therefore we cannot expect Cynnau|Ignite's benefits to be consistent across groups. Analytically controlling for this issue will be challenging because recruitment will take place at different times for each group and there is no guarantee that participation and recruitment quotas will be met, therefore it will not be possible to create a matched control cohort based on career pathway. While the addition of a control group is itself beneficial, an unmatched control group presents more of a challenge separating the direct impact of Cynnau|Ignite from wider changes in the sector, happening at Cardiff University and throughout the UK during 2024/25. Additionally, several of the measures described in this study were purposely built for the study and are self-reported, which limits the validity of findings. However, the use of mixed methods, multi assessments, and multi-informants should allow for enough data triangulation to ensure overall validity. Despite these limitations, we feel this is a robust theory-driven evaluation which is well placed to determine the effectiveness of Cynnau|Ignite and its feasibility for use in promoting PRC practices at Cardiff University and further afield.

## Ethics and dissemination

### Consent

All participants must provide informed consent to take part in the evaluation. They will be advised that they have the right to withdraw from the study at any point without penalty, and may withdraw their data up to five working days after its submission. As Cardiff University is an Open Access University, participants will also be invited to consent for their de-identified data to be made openly accessible for the wider scientific community in the UK Data Service repository (https://ukdataservice.ac.uk) and general public, and for direct quotes and excerpts of their data to be used for research dissemination and educational purposes.

### Confidentiality

Data will remain confidential throughout this research. Participation is not anonymous and all participants will be asked to provide their email address each time they are asked to provide data. A key linking participants' email address to an ID number will be created. In this way, personal data can be stored separately from other research data, linked via participants' ID number. All interviews will be transcribed within 72 hours and entered to NVivo, where they

will be stored anonymously under the participant ID number. Participant responses will be anonymised and stored under a participant number only, on an encrypted and password-protected computer and university servers. In line with GDPR guidelines, personal data will be retained until completion of the project (the time specified above - 01/02/26). Consent forms will be retained until 5yrs after completion of the project.

## Harms

We do not anticipate any harm arising from participation in either the Cynnau|Ignite training programme or the control group but the process evaluation will enquire about negative outputs, experiences, and harms.

## Auditing

An independent Cynnau|Ignite Advisory Panel will be consulted for quality assurance at every stage of the programme, including the evaluation detailed in this protocol. This group will consist of individuals with expertise in the design and evaluation of organisational training programs and change interventions, to ensure Cynnau|Ignite and its evaluation meet the expectations of participants and stakeholders.

## Access to data

Only the research team will have access to primary data. When shared beyond the research team, all data will be stored under participant numbers only and will be fully de-identified and anonymised.

## Dissemination policy

These results will be disseminated throughout Cardiff University in the form of reports and presentations targeted towards programme stakeholders, in particular senior management, heads of school, and staff. Recommendations regarding future implementation of Cynnau|Ignite will be shared with the design team and senior management. Beyond the university, results from this study may be published or presented in academic journals, conference publications, and other engagement and dissemination events.

## Supporting Information

**S1 File. Module Learning Outcomes.**
(DOCX)

**S2 File. Cynnau|Ignite Evaluation Materials Table.**
(DOCX)

## Acknowledgments

The authors wish to thank Professor Roger Whitaker, for helping to make this project possible through his contribution to funding acquisition and supervision. With thanks to Dr Simon Murphy for reviewing this protocol, as well as Cardiff University for their ongoing support of Cynnau|Ignite and its evaluation.

## Author contributions

**Conceptualization:** Charlotte Hennah, Nicola Edwards, Hayley Beckett, James Vilares, Karin Wahl-Jorgensen, Sofia Gameiro.

**Data curation:** Sofia Gameiro.

**Funding acquisition:** Nicola Edwards, James Vilares, Karin Wahl-Jorgensen, Sofia Gameiro.

**Investigation:** Sofia Gameiro.

**Methodology:** Charlotte Hennah, Maleeha Rizwan, Nicola Edwards, Hayley Beckett, Heidi Dawson, Jonathan Morris, James Vilares, Sofia Gameiro.

**Project administration:** Michele Convery, Heidi Dawson, Mair Rigby, Karin Wahl-Jorgensen, Rebecca Williams.

**Resources:** Heidi Dawson.

**Supervision:** Nicola Edwards, Heidi Dawson, James Vilares, Karin Wahl-Jorgensen, Sofia Gameiro.

**Visualization:** Charlotte Hennah, Maleeha Rizwan, Heidi Dawson, Mair Rigby, Sofia Gameiro.

**Writing – original draft:** Charlotte Hennah, Maleeha Rizwan, Sofia Gameiro.

**Writing – review & editing:** Maleeha Rizwan, Nicola Edwards, Hayley Beckett, Michele Convery, Heidi Dawson, Jonathan Morris, Mair Rigby, James Vilares, Karin Wahl-Jorgensen, Rebecca Williams, Sofia Gameiro.

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
