## [Decision Letter · Decision Letter 0]

26 Jan 2025

Protocol for a theory-based, mixed methods evaluation of Cynnau|Ignite: an active learning programme to foster positive research culture through leadership development at Cardiff University.

PONE-D-24-55112

Dear Dr. Charlotte Hennah,

We’re pleased to inform you that your manuscript has been judged scientifically suitable for publication and will be formally accepted for publication once it meets all outstanding technical requirements.

Kind regards,

Agnieszka Stachowiak, Ph.D., Eng., Asc. Prof.

Academic Editor

PLOS ONE

1. Your ethics statement should only appear in the Methods section of your manuscript. If your ethics statement is written in any section besides the Methods, please move it to the Methods section and delete it from any other section. Please ensure that your ethics statement is included in your manuscript, as the ethics statement entered into the online submission form will not be published alongside your manuscript.

Additional Editor Comments (optional):

Dear Author,

congratulations on your submission,

there are some minor improvement recommended, please read the review carefully and use it to improve your future papers,

regards,

Editor

Reviewers' comments:

Reviewer's Responses to Questions

**Comments to the Author**

1. Does the manuscript provide a valid rationale for the proposed study, with clearly identified and justified research questions?

Reviewer #1: Yes

Reviewer #2: Yes

2. Is the protocol technically sound and planned in a manner that will lead to a meaningful outcome and allow testing the stated hypotheses?

Reviewer #1: Partly

Reviewer #2: Yes

3. Is the methodology feasible and described in sufficient detail to allow the work to be replicable?

Reviewer #1: Yes

Reviewer #2: Yes

4. Have the authors described where all data underlying the findings will be made available when the study is complete?

Reviewer #1: Yes

Reviewer #2: Yes

5. Is the manuscript presented in an intelligible fashion and written in standard English?

Reviewer #1: Yes

Reviewer #2: Yes

6. Review Comments to the Author

You may also provide optional suggestions and comments to authors that they might find helpful in planning their study.

Reviewer #1: This paper describes an intervention in Cardiff University to foster and support PRC with a mixed methods approach to determine if the intervention is successful. The intervention is sound and the outcomes will be informative to others looking to also enhance their culture. I would suggest that the study design include additional follow up later on to see if the interventions were successful in changing the culture of the institution and not just a short term follow up.

Reviewer #2: The article fully described the proposed method for implementing the Cynnau|Ignite training and research program.

7. PLOS authors have the option to publish the peer review history of their article (what does this mean? ). If published, this will include your full peer review and any attached files.

**Do you want your identity to be public for this peer review?** For information about this choice, including consent withdrawal, please see our Privacy Policy .

Reviewer #1: No

Reviewer #2: No

---

## [Editor Report · Acceptance letter]

PONE-D-24-55112

PLOS ONE

Dear Dr. Hennah,

I'm pleased to inform you that your manuscript has been deemed suitable for publication in PLOS ONE. Congratulations! Your manuscript is now being handed over to our production team.

Kind regards,

on behalf of

Dr. Agnieszka Stachowiak

Academic Editor

PLOS ONE